# Myoinvasive Pattern as a Prognostic Marker in Low-Grade, Early-Stage Endometrioid Endometrial Carcinoma

**DOI:** 10.3390/cancers11121845

**Published:** 2019-11-22

**Authors:** Ignacio Ruz-Caracuel, Jorge L Ramón-Patino, Álvaro López-Janeiro, Laura Yébenes, Alberto Berjón, Alicia Hernández, Alejandro Gallego, Victoria Heredia-Soto, Marta Mendiola, Andrés Redondo, Alberto Peláez-García, David Hardisson

**Affiliations:** 1Department of Pathology, Hospital Universitario La Paz, IdiPAZ, 28046 Madrid, Spain; 2Department of Medical Oncology, Hospital Universitario La Paz, IdiPAZ, 28046 Madrid, Spain; 3Molecular Pathology and Therapeutic Targets Group, La Paz University Hospital (IdiPAZ), 28046 Madrid, Spain; 4Department of Obstetrics & Gynaecology, Hospital Universitario La Paz, IdiPAZ, 28046 Madrid, Spain; 5Translational Oncology Research Laboratory, La Paz University Hospital (IdiPAZ), 28046 Madrid, Spain; 6Center for Biomedical Research in the Cancer Network (Centro de Investigación Biomédica en Red de Cáncer, CIBERONC), Instituto de Salud Carlos III, 28046 Madrid, Spain; 7Faculty of Medicine, Universidad Autónoma de Madrid, 28046 Madrid, Spain

**Keywords:** endometrial carcinoma, myoinvasive pattern, prognosis, microsatellite instability, mismatch repair protein, early stage

## Abstract

Low-grade and early Federation for Gynecology and Obstetrics (FIGO) stage endometrioid endometrial carcinomas (EEC) have an excellent prognosis. However, approximately 10% of patients develop recurrence, which cannot be correctly predicted at diagnosis. We evaluated myoinvasive patterns as a prognostic factor of relapse in low-grade, early-stage EEC. Two-hundred and fifty-eight cases were selected according to the following inclusion criteria: (i) endometrioid endometrial carcinomas, (ii) grade 1 or 2 with (iii) FIGO stage I or II, and (iv) clinical follow-up. Slides were reviewed to annotate the myoinvasive pattern present in each case (infiltrative glands, microcystic, elongated and fragmented –MELF-, broad front, adenomyosis-like and adenoma malignum). Microsatellite instability was studied by immunoexpression of mismatch repair proteins (MLH1, PMS2, MSH2, and MSH6). There were 29 recurrences (11.2%) among the 258 cases analysed. A predominant broad front myoinvasive pattern was significantly associated with tumour relapse (*p* = 0.003). The presence of a pattern of infiltrative glands (*p* = 0.001) and microsatellite instability (*p* = 0.004) were associated with lower disease-free survival, without having an impact on overall survival. Our observations suggest the potential value of the pattern of myoinvasion as a prognostic factor in low-grade, early-stage endometrioid endometrial carcinoma.

Presented in part at the 31st European Congress of Pathology, Nice (France), 7–11 September 2019 (OFTP-14-001).

## 1. Introduction

Endometrial cancer is the most common malignant tumour of the female genital tract [1]. Moreover, its incidence is increasing in the Western world in a probable relation with a greater prevalence of obesity and metabolic syndromes in these regions, in addition to the ageing of the population [2]. Nearly 80% of endometrial cancers are diagnosed at an early stage (International Federation of Obstetrics and Gynaecology–FIGO-stages I or II) with a 5-year overall survival ranging from 74 to 91% [3]. However, there are a subset of patients that range from 4.4% in stage IA to 13.7% in stage II that experience tumour recurrence [4], mainly in the first three years after surgery [5].

Although classical parameters associated with a higher risk of recurrence (increasing age, depth of myometrial invasion, histological tumour type and grade, presence of lymphovascular invasion, and FIGO stage) have been included in several risk stratification systems, they have not demonstrated enough accuracy to stratify recurrence risk in early stage endometrial cancer [6]. Using these systems, low-grade (grade 1 or 2) and early-stage endometrioid endometrial carcinomas (EEC) are classified in low or intermediate-low risk groups. For this reason, there is a need to find new parameters to identify patients with higher risk of relapse among those diagnosed at an early stage in order to treat or follow them up properly without overtreating patients with an excellent prognosis. This has been identified as one of the main improvements in endometrial cancer classification and staging that needs to be addressed in the near future [7].

Since the proper recognition of five patterns of EEC myoinvasion [8], much attention has been paid to the microcystic, elongated and fragmented glands (MELF) pattern of myoinvasion. The MELF pattern has been associated with an increased risk of lymph node metastases, but without any impact on prognosis [9,10]. In contrast, there is little published data regarding the risk of recurrence associated with any of the other patterns [11,12].

The present study investigated the association between patterns of myoinvasion and recurrence in low-grade, early-stage endometrial endometrioid cancer. Because these tumours are confined to the uterus, the local microenvironment must play a role in the behaviour of these malignancies. One of the local microenvironment factors is the interaction with the tumour-stroma [13]. We hypothesised that the patterns of myoinvasion could be a reflection of this interaction.

To our knowledge, there are no studies examining the relationship between microsatellite instability (MSI) and myoinvasive patterns in EEC. Moreover, the role of MSI and its prognostic value has not been clearly established in early-stage EEC [14,15,16]. Since mismatch repair protein (MMRp) immunoexpression assessment is routinely performed in clinical practice in Lynch syndrome screening [17], it appears to be interesting to evaluate whether this biomarker could add relevant information of prognostic value in early stage EEC.

In this study, we describe the distribution of the five infiltrative patterns in a series of low-grade, early-stage EEC. We analysed the value of myoinvasion patterns and MSI as prognostic factors of relapse in this tumour.

## 2. Results

### 2.1. Clinicopathological Characteristics

A total of 512 endometrial cancer patients underwent a hysterectomy and bilateral salpingo-oophorectomy between January 2003 and December 2015. Among these, there were 258 eligible cases of low-grade, early-stage EEC that were treated in the same centre, which were included in our study. A total of 29 tumour recurrences were detected during the follow-up period.

Clinicopathological parameters of the tumours distributed in recurrence and non-recurrence groups are presented in Table 1. The mean age of patients at the time of diagnosis was 63.96 years (standard deviation, SD, 10.31; range, 37–88). The frequencies of patients in each stage were as follows: 182 (70.5%) FIGO Stage IA, 67 (26%) FIGO Stage IB, and 9 (3.5%) FIGO Stage II. There were 56 (21.7%) EEC grade 2 in opposition to 201 (77.9%) EEC grade 1. Thirty-nine (15.1%) tumours had lymphovascular invasion.

All five myoinvasive patterns were present in our series (Figure 1). The most frequent predominant pattern of myoinvasion was the infiltrative glands (50.39%), followed by the broad front (10.47%), and the adenomyosis-like (10.08%). Eighteen cases (6.98%) showed a predominant MELF myoinvasive pattern, and only one case (0.39%) showed the so-called adenoma malignum pattern. Moreover, there were 56 (21.71%) non-infiltrative EEC, all of them in the non-recurrence group (Table 1). Mucinous and squamous differentiation was also assessed (Table 1). In total, there were 64 cases (24.81%) with mucinous differentiation, and 91 cases (35.30%) with squamous differentiation.

Classical clinicopathological prognostic factors that demonstrated a significant association with relapse in our series were age (*p* = 0.039), FIGO stage (*p* < 0.001), tumour grade (*p* = 0.029), depth of myometrial invasion (<0.001), and presence of lymphovascular space invasion (0.001) (Table 1).

Association between myoinvasive predominant patterns and relapse showed that only the broad front predominant pattern was significantly associated with relapse (*p* = 0.003) (Table 1). Moreover, tumour relapse was significantly associated with the presence of any percentage of infiltrative glands at the myoinvasive front of the tumour (*p* = 0.034) (Table 1).

### 2.2. Myoinvasive Patterns and Microsatellite Instability

Immunoexpression of MMRp showed 31 (15.58%) unstable tumours out of 199 evaluable cases, with loss of expression of at least one of the four proteins. These tumours were considered microsatellite unstable. The cases had the following distribution: 19 (61.3%) with loss of MLH1/PMS2, 3 (9.7%) with loss of MSH2/MSH6, 4 (12.9%) with an isolated loss of PMS2, and 5 (16.1%) with an isolated loss of MSH6. *MLH1* promoter methylation was detected in all cases with loss of MLH1 protein expression. The MSI phenotype was statistically associated with tumour relapse (*p* = 0.030).

The presence of infiltrative glands at the myoinvasive front was the only parameter significantly associated with MSI (*p* = 0.019) (Table 2). Of note, only the presence of the MELF pattern was associated with both, mucinous and squamous differentiation (*p* = 0.001 and *p* = 0.009, respectively) (Table 2).

### 2.3. Prognosis

Classical pathological parameters such as grade 2 vs grade 1 (*p* = 0.029), FIGO Stage II or IB vs IA (*p* < 0.001), and presence of lymphovascular space invasion (*p* = 0.001) were associated with tumour relapse in the univariate analysis using a logistic regression model (Table 3). Moreover, the presence of the broad front predominant pattern (*p* = 0.001) and the presence of any percentage of infiltrative glands at the myoinvasive front (*p* = 0.040) were also significantly associated with tumour relapse. In contrast, MSI appeared as a protective factor for tumour relapse in our series (*p* = 0.014). A multivariate logistic regression model was done using the significant values (*p* < 0.05) from the univariate analysis, age ≥ 60 years was added because it is a known parameter associated with relapse [6]. In the multivariate analysis, only the broad front predominant pattern of myoinvasion appeared to be significantly associated with relapse (Table 4).

Median follow-up of patients in our series was 83.96 ± 43.39 months. Tumour relapse appeared at a median follow-up of 32.24 ± 25.91 months. In the whole series, disease-free survival (DFS) was 80.49 ± 45.59 months (minimum: 0 months; maximum: 174 months) and overall survival (OS) was 83.96 ± 43.39 months (minimum: 0; maximum: 174 months). Myoinvasion patterns and the presence of an infiltrative glands pattern showed a negative impact on DFS (*p* = 0.001 and *p* = 0.004, respectively) (Figure 2A,C), although they did not show any statistically significant difference on OS (Figure 2B,D). Cases with MSI had a lower DFS (*p* = 0.036), and showed a trend towards poorer OS (*p* = 0.129) (Figure 2E,F).

## 3. Discussion

Since the description of different myoinvasive patterns in EEC in 2013 by Cole and Quick [8], little attention has been paid to analyse the relation between these patterns and the outcome of the tumour [11,12]. To our knowledge, this is the first series correlating myoinvasive patterns with recurrence in low-grade, early-stage EEC. In fact, this study comprising a large series of morphologically well-characterized low-grade and early-stage EEC demonstrated that the broad front predominant myoinvasive pattern was associated with an increased risk of tumour relapse.

In agreement with previous results, the infiltrative glands pattern was the most frequent pattern of myoinvasion in our series, and is associated with other prognostic risk factors [12]. The remaining patterns were present in lower proportion. The adenoma malignum, also called minimal deviation endometrial carcinoma, is a rarity in most series [11].

In contrast to our results, Suzuki et al. [18] described that the infiltrative pattern was associated with a worse prognosis in their series in comparison with the so-called expansile pattern. However, these data cannot be directly extrapolated to our series because these authors used a different classification system to the one described in 2013, which is now widely accepted [8]. Moreover, they have few cases within the infiltrative category, in contrast to what is widely described [9,10,11,12]. In our study, we found an association between the presence of the infiltrative glands pattern and the DFS of the patients. Similarly, Park et al. [12] described an association between this myoinvasive pattern and disease relapse, although their series also included grade 3 tumours and higher-stage tumours.

We describe for the first time an association between the broad front predominant myoinvasive pattern and recurrence risk, which has an impact on the DFS of the patients. This association is independent of other prognostic parameters, such as lymphovascular space invasion, depth of myometrial invasion, and advanced age. Previous studies did not focus on the low-grade and low-stage EEC population (low-risk cases). A distinction between the non-infiltrative and the broad front myoinvasive pattern in EEC is a common problem in clinical practice [19,20], and prompted a change from a three-tiered FIGO staging system (IA–tumour limited to endometrium, IB–tumour invading less than half of the myometrium, IC–tumour invading one-half or more of the myometrium) to the current two-tiered (IA–tumour limited to endometrium or invading less than half of the myometrium, IB–tumour invading one-half or more of the myometrium) staging system several years ago. In our series, we applied very strict criteria to define the myoinvasive patterns in order to avoid misclassification of cases [8]. These criteria were the presence of a linear tumour front (in contrast to an undulating contour), the presence of desmoplasia and/or the existence of a “shoulder” of infiltrating glands (Figure 1E). Interestingly, none of the low-grade/non-infiltrative EEC showed tumour relapse, being a subpopulation with an excellent prognosis.

MELF pattern has been widely studied since its recognition by Murray at al. [21] in 2003 due to its association with lymph metastases [22,23,24,25]. The association between MELF pattern in grade 1–2 EEC and extrauterine disease has also been proven [26]. However, it seems that MELF pattern has little or no impact on prognosis [9,26,27]. Our results support this conclusion, since we did not find any association between MELF predominant pattern or the presence of MELF pattern and the risk of recurrence. We found a statistically significant association between the presence of MELF pattern with depth of myometrial invasion and the presence of lymphovascular invasion [28,29]. MELF pattern is also associated with mucinous and squamous differentiation [28], but not with MSI [30].

In our series of low-grade low-stage EEC 15.58% of cases were microsatellite unstable. This was lower to what it is generally described in EEC, because our study population only comprised grade 1 and 2 tumours, and MSI is usually associated with a higher histopathological grade [16]. Only the presence of the infiltrative glands myoinvasive pattern was significantly associated with MSI.

There are conflicting results regarding the prognostic value of MSI in endometrial carcinoma. If we extrapolate data from the TCGA program, most low-grade EEC fall within the MSI-hypermutated and the low copy number molecular subgroups. Both subgroups share a similar prognosis [31], being MSI the most relevant distinguishing factor. These results were in agreement with previous studies that did not find a prognostic value in terms of DFS or OS for MSI in EEC stages I to IV [16,32,33]. Of note, in early stage EEC some groups have identified MSI as an independent factor of tumour relapse [14] or survival [16]; whereas others have not found any association [15]. In the univariate analysis, we found a higher risk of recurrence for microsatellite unstable patients as well as an impact on DFS. However, this association was non-significant in the multivariate analysis, which may be explained by the association of MSI with other prognostic markers, as described in previous studies [33].

The main limitation of our study was the relative low number of tumour relapses, which may limit the conclusions of the study. Unfortunately, this a common problem in studies concerning pathologies with a low incidence of events, as is the case of EEC. On the other side, one of the strengths of this study is the homogeneity of the series, which includes only low-grade, early-stage EEC, together with the strict morphological criteria used to classify the different infiltration patterns of the tumours.

## 4. Materials and Methods

### 4.1. Patients Selection

Cases were retrieved from the archive of the Department of Pathology at the Hospital Universitario La Paz (Madrid, Spain). Selected cases had to fulfil the following requirements: primary endometrioid endometrial carcinoma grade 1 or 2, FIGO Stage I or II, hysterectomy and bilateral salpingo-oophorectomy performed between January 2003 and December 2015, and a follow-up of at least five years. Histological type was considered according to the 2014 World Health Organization classification of tumours of the endometrium [34]. Patients were staged according to the International Federation for Gynecology and Obstetrics (FIGO) classification [35]. Prognostic factors such as depth of infiltration, lymphovascular space invasion, and tumour grade were retrieved from pathological reports and reassessed during pathological evaluation of myoinvasive patterns. Clinical data were taken from medical oncology clinical reports.

This study was approved by the Ethics Committee of the University Hospital La Paz, Madrid, Spain (code HULP: PI-3108), and was conducted in accordance with ethical standards of the Helsinki Declaration of the World Medical Association.

### 4.2. Histopathological Analysis

For each case, all haematoxylin and eosin (HE)-stained sections (number of slides 1–24; mean: 6.66 per case) were re-examined, and the following variables were recorded: the patterns of myoinvasion present, the most frequent pattern of myoinvasion, the presence of squamous differentiation, and the presence of mucinous differentiation. Myoinvasive patterns were evaluated according to the classification described by Cole and Quick [8], and classified into six groups: non-invasive, infiltrative glands, broad front, adenomyosis-like, microcystic, elongated and fragmented glands (MELF), and adenoma malignum. Lymphovascular invasion was reassessed in cases with MELF pattern of myoinvasion.

### 4.3. Tissue Microarray Construction and Immunohistochemistry of DNA Mismatch Repair Proteins

Representative non-necrotic central areas of each tumour were marked on H&E slides. Two representative cores of 1.2 mm of diameter were taken from the selected areas of the paraffin block. The tissue cores were arrayed into a receptor paraffin block using a tissue microarray (TMA) workstation (Beecher Instruments, Silver Spring, MD, USA) as described previously [36].

Immunohistochemistry was performed on 4 μm sections of the TMA blocks using the following primary antibodies: MLH1 (clone ES05, Dako, Glostrup, Denmark; prediluted), PMS2 (clone EP51, Dako, prediluted), MSH2 (clone FE11, Dako, prediluted), MSH6 (clone EP49, Dako, prediluted). Staining was performed using an OMNIS autostainer (Dako) according to the manufacturer´s instructions.

Analysis of immunohistochemical stains was performed as described previously [36]. Whole slide immunohistochemistry was performed in doubtful cases.

### 4.4. Statistical Analysis

The chi-squared or Fisher’s exact test was used to evaluate the association between qualitative variables. Student *t*-test was used to evaluate the association between ages in both groups. DFS was defined as the time from the date of diagnosis to relapse or death due to any cause, and was determined using routine clinical practice criteria. OS was defined as the time from the date of diagnosis to death due to any cause.

Quantitative results were expressed as mean ± SD. Prognostic clinicopathological factors in recurrence and non-recurrence groups were statistically studied with univariate logistic regression. According to these results, multivariate logistic regression was modelled using the significant values (*p* < 0.05) from the univariate analysis and the age ≥ 60 years. DFS and OS data were plotted in Kaplan–Meier curves, and a long-rank test was used to compare these parameters. The data were analysed using statistical software IBM SPSS V19. Differences were considered significant with *p* values < 0.05.

## 5. Conclusions

In conclusion, we described for the first time an independent association between the predominant broad front myoinvasive pattern and tumour relapse in early-stage, low-grade EEC. We also described a significant association between the presence of infiltrative glands pattern and DFS. Predominant pattern of myoinvasion and the presence of infiltrative glands pattern did not have any impact on OS in our series. MSI was associated with lower DFS, without having an impact on OS in low-grade, early-stage EEC.

Our observations have suggested the potential value of the pattern of myoinvasion as a prognostic factor of relapse in low-grade, early-stage EEC.

## Figures and Tables

**Figure 1 cancers-11-01845-f001:**
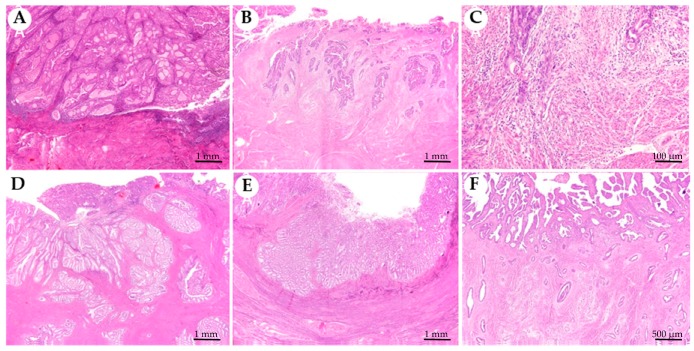
Myoinvasive patterns present in endometrioid endometrial carcinomas. (**A**) Non infiltrative, tumour is confined to the endometrium and does not invade the myometrium. (**B**) Infiltrative glands, composed of single or small groups of glands with irregular contours scattered in the myometrium. (**C**) Microcystic, elongated and fragmented glands (MELF) identified at the leading edge of the tumour associated with neutrophils infiltrate. (**D**) Adenomyosis-like, composed of groups larger than five glands with round borders within myometrium simulating adenomyosis. (**E**) Broad front, which has a linear limit between the tumour and the stroma beneath. (**F**) Adenoma malignum is composed of low cytological grade, regular, rounded, and widely spaced glands.

**Figure 2 cancers-11-01845-f002:**
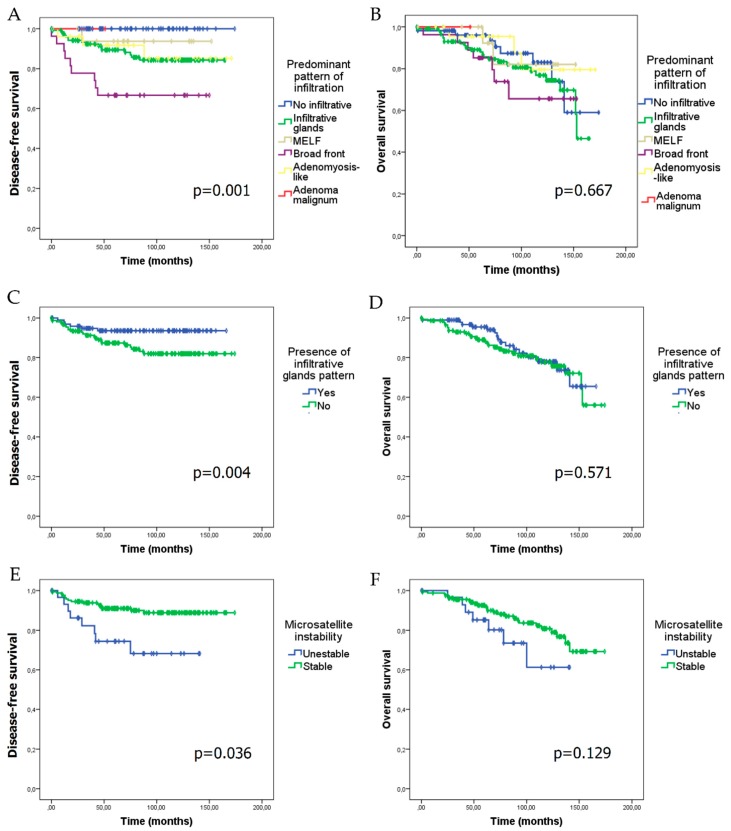
Kaplan–Meier curves for disease-free survival (DFS) and overall survival (OS) according to the predominant pattern of myoinvasion, the presence of infiltrative glands myoinvasive pattern, and microsatellite instability (MSI). DFS (**A**) and OS (**B**) in relation to the predominant myoinvasive pattern. DFS (**C**) and OS (**D**) according to the presence of the infiltrative glands myoinvasive pattern. DFS (**E**) and OS (**F**) in microsatellite unstable versus stable cases.

**Table 1 cancers-11-01845-t001:** Clinicopathological characteristics of the study population.

Parameter		Total (*n* = 258)	No Recurrence (*n* = 229)	Recurrence (*n* = 29)	*p*
**Age (mean)**		63.96 ± 10.31	63.54 ± 10.45	67.28 ± 8.64	0.039
**Age**					
<60 y.		94 (36.4%)	88 (38.4%)	6 (20.7%)	0.065
≥60 y.		164 (63.6%)	141 (61.6%)	23 (79.3%)	
**FIGO Stage**					
IA		182 (70.5%)	171 (74.7%)	11 (37.9%)	<0.001
IB		67 (26.0%)	51 (22.3%)	16 (55.2%)	
II		9 (3.5%)	7 (3.1%)	2 (6.9%)	
**Tumour Grade**					
G1		201 (77.9%)	183 (80.3%)	18 (62.1%)	0.029
G2		56 (21.7%)	45 (19.7%)	11 (37.9%)	
**Depth of Myometrial Invasion**					
None		56 (21.71%)	56 (24.5%)	0	<0.001
Inner half		131 50.76%)	120 (52.4%)	11 (37.9%)	
Outer half		71 (27.5%)	53 (23.1%)	18 (62.1%)	
**Lymphovascular Space Invasion**					
Present		39 (15.1%)	28 (12.2%)	11 (37.9%)	0.001
Absent		219 (84.9%)	201 (87.8%)	18 (62.1%)	
**Mucinous Differentiation**					
Present		64 (24.8%)	53 (23.1%)	11 (37.9%)	0.085
Absent		194 (75.2%)	176 (76.9%)	18 (62.1%)	
**Squamous Differentiation**					
Present		91 (35.3%)	80 (34.9%)	11 (37.9%)	0.727
Absent		167 (64.7%)	149 (65.1%)	18 (62.1%)	
**Predominant Myoinvasive Pattern**					
Non infiltrative	Present	56 (21.7%)	56 (24.5%)	0	0.003
	Absent	202 (78.3%)	173 (75.5%)	29 (100%)	
Infiltrating glands	Present	130 (50.4%)	114 (49.8%)	16 (55.2%)	0.600
	Absent	128 (49.6%)	115 (50.2%)	13 (44.8%)	
MELF	Present	18 (7.0%)	17 (7.4%)	1 (3.4%)	0.703
	Absent	240 (93.0%)	212 (92.6%)	28 (96.6%)	
Broad front	Present	27 (10.5%)	18 (7.9%)	9 (31.0%)	0.003
	Absent	231 (89.5%)	211 (92.1%)	20 (69%)	
Adenomyosis-like	Present	26 (10.1%)	23 (10.0%)	3 (10.3%)	1.00
	Absent	232 (89.9%)	206 (90.0%)	26 (89.7%)	
A. malignum	Present	1 (0.4%)	1 (0.4%)	0	
	Absent	257 (99.6%)	228 (99.6%)	29 (100%)	
**Infiltrating Glands Pattern**					
Present		158 (61.2%)	135 (59.0%)	23 (79.3%)	0.036
Absent		100 (38.8%)	94 (41.0%)	6 (20.7%)	
**MELF Pattern**					
Present		30 (11.6%)	28 (12.2%)	2 (6.9%)	0.547
Absent		228 (88.4%)	201 (87.8%)	27 (93.1%)	
**Microsatellite Instability**					
Stable		168 (84.4%)	152 (86.9%)	16 (66.7%)	0.030
Unstable		31 (15.6%)	23 (13.1%)	8 (33.3%)	

FIGO, International Federation of Obstetrics and Gynaecology; MELF, microcystic, elongated and fragmented glands; A. malignum, adenoma malignum.

**Table 2 cancers-11-01845-t002:** Associations between myoinvasive patterns, microsatellite status, and morphological parameters.

Parameter	Microsatellite Instability	Mucinous Differentiation	Squamous Differentiation
Total number of cases analysed	199	258	258
Total number of positive cases (%)	31 (15.58)	64 (24.8)	91 (35.3)
Infiltrative glands predominant pattern	19 (*p* = 0.224)	34 (*p* = 0.613)	50 (*p* = 0.280)
MELF predominant pattern	1 (*p* = 0.481)	8 (*p* = 0.084)	9 (*p* = 0.175)
Broad front predominant pattern	4 (*p* = 0.311)	5 (*p* = 0.558)	11 (*p* = 0.336)
Adenomyosis-like predominant pattern	4 (*p* = 0.506)	8 (*p* = 0.458)	8 (*p* = 0.612)
Presence of infiltrative glands pattern	25 (*p* = 0.019)	43 (*p* = 0.260)	65 (*p* = 0.013)
Presence of MELF pattern	2 (*p* = 0.265)	15 (*p* = 0.001)	17 (*p* = 0.009)

**Table 3 cancers-11-01845-t003:** Univariate analyses of odds ratios in the logistic regression model with relapse as the dependent variable.

Parameter	Odds Ratio (CI 95%)	*p*-Value
**Univariate Logistic Regression**		
Age ≥ 60 years	2.392 (0.937–6.107)	0.068
Tumour grade 2	2.485 (1.097–5.631)	0.029
FIGO stage ≥IB	3.065 (1.658–5.667)	<0.001
Lymphovascular space invasion	4.387 (1.879–10.242)	0.001
Infiltrative glands predominant pattern	1.242 (0.571–2.699)	0.585
MELF predominant pattern	0.445 (0.057–3.477)	0.440
Broad front predominant pattern	4.751 (1.833–12.314)	0.001
Adenomyosis-like predominant pattern	1.033 (0.290–3.681)	0.960
Presence of infiltrative glands pattern	2.669 (1.047–6.807)	0.040
Presence of MELF pattern	0.532 (0.120–2.359)	0.406
Mucinous differentiation	2.029 (0.902–4.564)	0.087
Squamous differentiation	1.138 (0.513–2.527)	0.750
Microsatellite instability	3.304 (1.271–8.589)	0.014

**Table 4 cancers-11-01845-t004:** Multivariate analyses of odds ratios in the logistic regression model with relapse as the dependent variable.

Parameter	Odds Ratio (CI 95%)	*p*-Value
**Multivariate Logistic Regression**		
Age ≥ 60 years	2.772 (0.722–10.641)	0.137
Tumour grade 2	2.103 (0.700–6.324)	0.186
FIGO Stage ≥IB	1.722 (0.161–18.383)	0.653
Lymphovascular space invasion	2.729 (0.884–8.424)	0.081
Broad front predominant pattern	17.302 (3.482–85.985)	<0.001
Presence of infiltrative glands pattern	1.981 (0.497–7.895)	0.333
Microsatellite instability	1.453 (0.422–4.996)	0.553

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
