# Peer review of "Myoinvasive Pattern as a Prognostic Marker in Low-Grade, Early-Stage Endometrioid Endometrial Carcinoma"

_cancers, 2019, doi:10.3390/cancers11121845_

Round 1
Reviewer 1 Report
The authors evaluated myoinvasive patterns as a prognostic factor of relapse 28 in low-grade, early-stage EEC out of 258 cases with 5 criteria and studied microsatellite instability., since there is a subset of patients experience tumor recurrence. Multivariate analyses showed that only broad front predominant pattern was associated with tumor relapse. The presence of infiltrative glands pattern and microsatellite instability were associated with lower disease-free survival by Kaplan-Meier curves. They concluded that the potential value of the pattern of myoinvasion as a prognostic factor in low-grade, early-stage endometrioid endometrial carcinoma.
Just minor concerns
Figure 2 is not clear. Please improve the quality of the data.
Author Response
Response to Reviewer 1 Comments
The authors evaluated myoinvasive patterns as a prognostic factor of relapse 28 in low-grade, early-stage EEC out of 258 cases with 5 criteria and studied microsatellite instability., since there is a subset of patients experience tumor recurrence. Multivariate analyses showed that only broad front predominant pattern was associated with tumor relapse. The presence of infiltrative glands pattern and microsatellite instability were associated with lower disease-free survival by Kaplan-Meier curves. They concluded that the potential value of the pattern of myoinvasion as a prognostic factor in low-grade, early-stage endometrioid endometrial carcinoma.
Point 1: Figure 2 is not clear. Please improve the quality of the data.
Response 1: The quality of Figure 2 has been improved so data are now clearly legible.

Reviewer 2 Report
This is human cancer study based on clinicopathological characteristics, incl. predominant myoinvasive pattern and also immunoexpression data of mismatch repair proteins for prognosis (recurrence and disease-free survival) of early-stage endometrioid endometrial carcinoma. The study is of interest, because of its novelty and interesting findings. Nevertheless, it needs important corrections and clarifications regarding applied statistics. Statistical evaluation should be described in efficient way with explanation of application for specific study assumptions, eg. univariate and multivariate (what variables?) approaches for estimation of recurrence ORs, statistics description in tables. Estimation of p-values (Table 1) in specific features of predominant myoinvasive pattern is difficult to follow.
Minor remarks
Recurrence and DFS time (median, min, max, etc.) should be presented. 4.4. Section with DNA isolation and methylation of mismatch repair genes is unnecessary.
Author Response
Response to Reviewer 2 Comments
This is human cancer study based on clinicopathological characteristics, incl. predominant myoinvasive pattern and also immunoexpression data of mismatch repair proteins for prognosis (recurrence and disease-free survival) of early-stage endometrioid endometrial carcinoma. The study is of interest, because of its novelty and interesting findings. Nevertheless, it needs important corrections and clarifications regarding applied statistics
Point 1: Statistical evaluation should be described in efficient way with explanation of application for specific study assumptions, eg. univariate and multivariate (what variables?) approaches for estimation of recurrence ORs, statistics description in tables.
Response 1: Logistic regression was used to estimate recurrences. This has been now clarified in Tables 3 and 4. Details about the variables used in the multivariate logistic regression model have been included in the Results (Section 2.3, lines 140 and 143-145) and in the Material and Methods (Section 4.4., lines 271-274).
Point 2: Estimation of p-values (Table 1) in specific features of predominant myoinvasive pattern is difficult to follow.
Response 2: Table 1 has been modified according to the reviewer´s indications.
Point 3: Recurrence and DFS time (median, min, max, etc.) should be presented.
Response 3. These data have been now included in the text (Section 2.3, lines 155-157).
Point 4: Section with DNA isolation and methylation of mismatch repair genes is unnecessary.
Response 4: This section has been now deleted from the text.

Round 2
Reviewer 2 Report
The authors have adressed to all my comments.